# Integrating behavioral, social, and technological factors in cryptocurrency investment decisions

**Tang My Sang** *

Ho Chi Minh University of Banking, Ho Chi Minh City, Vietnam

* sangtm@hub.edu.vn

## Abstract

Cryptocurrency markets are characterized by high volatility and behavioral biases. Although several studies have examined psychological, social, and technological factors separately, few have comprehensively integrated them within a unified framework based on behavioral finance. This study addresses this gap by examining the interplay among digital financial literacy, impulsivity, social media financial influencers, fintech self-efficacy, and attitude toward investment in shaping investment decisions in Vietnam's cryptocurrency market. Data were collected from 505 individual investors, and partial least squares structural equation modeling was employed to test the proposed framework. The results indicate that digital financial literacy fosters positive attitudes toward investment and directly supports informed investment decision-making. However, contrary to conventional expectations, digital financial literacy is positively associated with impulsivity-related investment behavior, suggesting that higher perceived digital competence may foster overconfidence and an illusion of control in highly volatile cryptocurrency markets. Social media financial influencers significantly shape investors' attitudes and also exert a direct influence on investment decisions, highlighting both attitudinal and behavioral pathways. Furthermore, fintech self-efficacy moderates the relationship between attitudes and investment decisions by reducing investors' reliance on attitudinal cues when making investment choices. The findings highlight the importance of distinguishing rapid informed decision-making from affect-driven impulsivity and emphasize the role of overconfidence and perceived digital literacy in shaping investor behavior. Practically, the results call for targeted digital financial education and regulatory oversight of online financial content to promote informed and sustainable investment practices.

## 1. Introduction

Understanding investment behavior is essential to explain how individuals make financial decisions [1]. It also helps clarify how these behaviors influence market stability and sustainable financial development [2]. In the era of digital finance, this issue

**Data availability statement:** All relevant data are available at https://doi.org/10.6084/m9.figshare.30609314.

**Funding:** This research is funded by Ho Chi Minh University of Banking, Ho Chi Minh City, Vietnam (Grant document number: 40/QĐ-ĐHNH-HĐT, dated November 27, 2024; funding recipient: TMS). The funders had no role in study design, data collection and analysis, decision to publish, or preparation of the manuscript.

**Competing interests:** The author reports that there are no competing interests to declare.

becomes especially relevant in cryptocurrency markets. These markets are highly volatile, speculative, and often lack regulatory clarity [3–5]. Such characteristics make investor behavior a decisive factor for success or failure [6,7].

Behavioral finance theory explains that investors often deviate from rational decision-making. They are influenced by emotions, heuristics, and social interactions [8]. This tendency is stronger among young investors who dominate the cryptocurrency market but often lack experience and financial literacy [9]. Understanding their behavior is important for designing educational programs, financial tools, and policies that promote responsible investment [10].

Previous research has examined cryptocurrency investment behavior from several perspectives. Psychological studies emphasize individual biases [11]. Social research highlights the influence of others [12]. Technological studies explore how digital platforms shape investor actions [13]. Behavioral finance identifies biases such as herding [14], overconfidence [15], and impulsivity [16], all of which shape investment decisions. Social media financial influencers also affect investors' attitudes toward cryptocurrency [17]. FinTech platforms make transactions faster and easier, but also increase exposure to market risks [18]. Despite these findings, most studies analyze these factors separately. Few examine how behavioral, social, and technological elements interact in a single framework.

Recent research has combined the Theory of Planned Behavior and the Unified Theory of Acceptance and Use of Technology to explain technology adoption [19,20]. However, few studies have extended these frameworks by adding behavioral finance concepts. This study addresses this limitation by integrating impulsivity and fintech self-efficacy into the TPB and UTAUT model. The model connects both rational and irrational aspects of investor decision-making. The study also clarifies the mediating role of attitude toward investment, which remains inconsistent in cryptocurrency studies [21,22].

Vietnam offers a unique and meaningful context for this research. Cryptocurrency is not legally recognized as an asset or payment method. However, Vietnam ranks among the top countries in cryptocurrency adoption. Its collectivist culture, strong social media influence, and large young investor population make it an ideal setting to study behavioral and social dynamics in emerging digital markets [23,24].

Although previous studies on cryptocurrency investment behavior [9,11,25], including those conducted in Vietnam [26–28], have provided valuable insights, several critical gaps remain. The first gap is that most research has examined individual factors separately, such as herding behavior or perceived utility [28], without integrating behavioral, social, and technological determinants into a single framework. Emerging factors such as digital financial literacy, impulsivity, social media financial influencers, and fintech self-efficacy have not been studied together, which limits a comprehensive understanding of investment behavior in cryptocurrency markets [29].

The second gap concerns the mediating role of attitude toward investment, which is emphasized in the Theory of Planned Behavior as a central driver of behavior. In the cryptocurrency context, this role has not been tested clearly, and many studies have reported non-significant effects of attitude on investment outcomes [21,22].

Clarifying this mediating mechanism is necessary to refine and recalibrate the application of the theory in digital financial environments [30]. The third gap is related to fintech self-efficacy, a construct that reflects individuals' confidence in using financial technologies [31]. Very few studies have investigated this factor in cryptocurrency investment [19], especially in emerging markets. In Vietnam, where Millennials and Generation Z account for the majority of investors but often lack financial knowledge and face an ambiguous regulatory framework, exploring fintech self-efficacy becomes essential. This line of inquiry not only contributes to the extension of self-efficacy theory but also provides practical value for investor education, protection, and policy development [32].

This study was conducted with the main objective of exploring the factors affecting investment behavior, focusing on four important variables, including digital financial literacy, impulsivity, social media financial influencers, and fintech self-efficacy. The study also tested the mediating role of attitude toward investment in the relationship between the above factors and investment decision. By integrating behavioral, technological, and social aspects into a comprehensive model, the study aims to fill the remaining gaps in the theory and practice of cryptocurrency investment in developing markets. In terms of structure, the study includes an introduction, a literature review and theoretical basis, model and hypothesis development, research methodology, analysis of results, discussion, and finally a conclusion with managerial implications.

## 2. Literature review

### 2.1. Underpinning theory

This study is grounded in four fundamental theories, including the Unified Theory of Acceptance and Use of Technology (UTAUT), the Theory of Planned Behavior (TPB), Self-efficacy Theory, and Behavioral Finance Theory. These theoretical foundations are summarized and presented in Table 1 to provide a comprehensive framework for explaining cryptocurrency investment behavior.

The UTAUT provides a useful lens for explaining how social influence and technological conditions shape investment behavior [33]. In this study, it is applied to clarify how social media financial influencers affect attitudes toward cryptocurrency and subsequently guide investment decisions [34]. The TPB highlights the pivotal role of attitude in shaping behavior, which is particularly relevant for examining how digital financial literacy fosters positive investment attitudes and translates into actual decision-making [35,36]. Self-efficacy theory offers further explanatory power by emphasizing that confidence in using financial technologies, captured through fintech self-efficacy [36], can strengthen the link between attitude and investment decisions [34].

**Table 1. Summary of underpinning theories.**

| Theory | Key contribution | Limitation in cryptocurrency context | Source |
|---|---|---|---|
| The Unified theory of acceptance and use of technology (UTAUT) | Explains how social media financial influencers shape attitudes toward investment, which in turn leads to investment behavior. | Lacks psychological and individual capability factors such as impulsivity and self-efficacy. | [33,19,34] |
| Theory of Planned Behavior (TPB) | Explains that digital financial literacy enhances positive attitudes toward investment and drives investment decisions. Identifies attitude toward investment as a mediating factor shaping investment behavior | Overlooks technological enablers and emotional dimensions affecting investors' behavior in digital contexts. | [35,36] |
| Self-efficacy theory | Adds that fintech self-efficacy moderates the impact of attitudes toward investment on investment behavior, reflecting confidence in using technology. | Focuses mainly on individual confidence while neglecting social and emotional influences. | [37,38] |
| Behavioral finance theory | Identifies impulsivity as a behavioral bias that makes investors prone to rash or emotional decisions. | Explains impulsive and emotional behavior but omits cognitive and technological factors. | [39,16] |

Within the Behavioral Finance framework, investors' decisions are influenced by a wide range of cognitive and emotional biases. Overconfidence may lead investors to overestimate their knowledge or predictive ability in volatile cryptocurrency markets. Herding reflects the tendency to follow collective market sentiment, especially in online communities. Anchoring occurs when investors rely too heavily on initial price information or past price peaks. Loss aversion makes individuals more sensitive to potential losses than equivalent gains, resulting in defensive or reactive trading. Recognizing this broader set of biases provides a more complete theoretical foundation for understanding cryptocurrency investment behavior. In this study, impulsivity is viewed as one specific bias within this framework, representing a fast and emotion-driven response that becomes particularly salient in high-volatility digital environments.

The integration of UTAUT, TPB, Self-efficacy, and Behavioral Finance theories provides a comprehensive lens for understanding cryptocurrency investment behavior. UTAUT and TPB explain the rational and social components of behavioral intention. Self-efficacy adds personal capability as a moderating factor, while Behavioral Finance captures impulsive and emotional tendencies that deviate from rational decision-making. Together, these theories complement each other by bridging technological readiness, behavioral intention, psychological confidence, and emotional bias in one coherent framework.

## 2.2. Research hypothesis

Behavioral finance research indicates that limited financial literacy increases the likelihood of rash decisions, particularly in fast-paced trading environments such as cryptocurrencies [40]. Digital financial literacy extends beyond the ability to use financial technology tools [41]. It also reflects the capacity to analyze risks and anticipate long-term outcomes of investment choices [42].

Impulsivity represents a behavioral bias that can override rational analysis and lead to short-term or speculative investment decisions, particularly in volatile markets like cryptocurrency [43]. It is defined as the tendency to make quick decisions without fully considering the consequences [44]. Behavioral finance theory identifies impulsivity as a factor that can reduce the quality of financial decisions [45]. In this context, digital financial literacy acts as a protective mechanism, helping investors manage impulsive tendencies during the investment process [46]. Empirical studies show that individuals with higher levels of financial literacy are better at evaluating risks and returns, which reduces the appeal of short-term gains and curbs impulsive behavior [47,48]. Conversely, investors with lower financial understanding are more susceptible to emotional responses and external cues [49]. Based on this rationale, the following research hypothesis is proposed:

Hypothesis H1: Digital financial literacy has a negative impact on impulsivity.

Investment decisions reflect the ability of individuals to balance risk and return in their financial choices [50]. In the volatile cryptocurrency market, digital financial literacy plays an important role by enabling investors to use digital tools to access, process, and interpret market information [51]. However, unlike traditional financial markets, cryptocurrency markets are characterized by weak fundamentals, high volatility, and a dominance of retail investors, which limits the relevance of conventional fundamental or cyclical analysis [52].

In this context, digital financial literacy motivates cryptocurrency investment not through precise valuation, but through enhanced participation capability, risk awareness, and perceived behavioral control [11]. Financially literate investors are better equipped to understand market volatility, manage trading platforms, and engage with cryptocurrencies despite high uncertainty. Several studies indicate that digital financial literacy plays an important role in enhancing the quality of investment decisions among young Indonesians [53,54].

Recent evidence suggests that informed cryptocurrency investors emphasize risk management through hedging and diversification rather than relying on long-term fundamental valuation, indicating that financial literacy supports informed market participation under conditions of extreme volatility [55]. In line with the TPB, this competence strengthens perceived control and translates rational understanding into concrete behavior [35]. Based on the above arguments and

evidence, higher financial literacy enables investors to make more appropriate and consistent decisions aligned with long-term goals.

Hypothesis H2: Digital financial literacy has a positive impact on investment decisions.

Attitude toward investment shows how positively or negatively individuals perceive participation in the market [56]. This attitude can be shaped by perceptions that investing is a good idea, a practical option, or a preferred form of digital asset ownership [57]. Financial knowledge plays a crucial role in forming such attitudes, as it enables investors to assess risks and returns more objectively and to recognize the potential for long-term value accumulation despite market volatility [58]. Empirical research in India and Indonesia has demonstrated that digital financial literacy is significantly correlated with financial attitudes, emphasizing the role of knowledge in shaping positive views toward investment decisions [8,53,59]. According to the TPB, attitudes are a central determinant of behavior [35]. Therefore, it can be inferred that improved digital financial literacy contributes to more positive attitudes toward cryptocurrency investment. The hypothesis is then proposed:

Hypothesis H3: Digital financial literacy has a positive influence on attitude toward investment.

In the digital age, a significant portion of investors, especially the younger generation, shape their financial attitudes through social media financial influencers [29]. These are individuals who are followed for their expertise, up-to-date information, credibility, and credibility in sharing financial information and investment opportunities [57]. Rather than analyzing technical data or complex reports themselves, many investors receive market signals and investment opinions by "following and listening" to influencers whom they consider to be experts, up to date, trustworthy, and reliable [60]. This suggests that attitudes toward investing in cryptocurrencies are largely influenced by social media rather than personal analysis. The UTAUT emphasizes the important role of social influence in shaping attitudes and behaviors, explaining how influencers can promote trust and positive attitudes toward investing [61]. Recent studies have also shown that the credibility and perceived relevance of social media financial influencers have a strong impact on willingness to participate in the cryptocurrency market [29]. Therefore, this study proposes:

Hypothesis H4: Social-media financial influencers have a positive influence on attitudes towards investment.

Impulsivity in the financial context is understood as the tendency to make quick, thoughtless decisions based on emotions or immediate impulses rather than rational analysis [62]. The cryptocurrency market's volatility, speculative appeal, and constant media exposure may amplify impulsive tendencies, making investors act on short-term excitement rather than long-term evaluation [63]. In cryptocurrency investing, impulsivity is often expressed by individuals investing immediately when they see an opportunity, acting on the spur of the moment, or trading without much consideration.

According to Behavioral Finance Theory, impulsivity is a behavioral bias that reduces rationality, making decisions emotional and accepting high risks [62]. Behavioral mechanisms such as sensation seeking and delay discounting make impulsivity particularly relevant in the context of cryptocurrency investment. Previous studies have shown that impulsivity is closely related to risky financial behavior. Frigerio et al. found that impulsive individuals are more likely to make financial mistakes because they overlook long-term analysis [64]. Similarly, research on cryptocurrency investing has highlighted the impact of the FOMO (Fear of Missing Out) effect, which drives investors to act hastily [20]. Loss of emotional control and the desire for quick profits obscure cost-benefit analysis, making impulsivity a key predictor of cryptocurrency investment behavior.

Hypothesis H5: Impulsivity has a positive influence on investment decisions.

In the framework of the TPB, attitudes are considered to be a core factor of actual behavior [35]. When individuals develop positive attitudes, they will form stronger intentions and more easily materialize them into investment decisions [65]. Previous studies have shown that positive attitudes towards investment play an important role in shaping investment behavior [21,65]. Investors who maintain favorable attitudes are more likely to believe in the long-term value of investing and are more likely to engage in financial decisions [66]. Experiments in India show that financial attitudes have a significant impact on stock investment intentions [67], while research in Indonesia confirms a positive association between

favorable attitudes and cryptocurrency investment behavior [68]. This implies that even with the same level of financial knowledge, individuals who maintain a positive attitude towards investment are more likely to make investment decisions. Consequently, the research puts forth the following hypothesis:

Hypothesis H6: Attitude towards investment has a positive influence on investment decisions.

Social influence goes beyond shaping attitudes and can also directly impact investment decisions [69]. In practice, many cryptocurrency investors often rely on direct recommendations from finfluencers without much consideration for their personal financial knowledge [17]. Trust, credibility, and the ability to spread information from social media reduce the need for personal analysis, thereby shortening the decision-making process [70]. Research results by Merkley et al. show that advice from influencers can strongly stimulate short-term capital inflows into cryptocurrencies, regardless of the fundamentals of the asset [71]. Similarly, research by Ozuem et al. demonstrates that digital social influence is directly linked to behavior among millennials [72]. This evidence is consistent with the argument in the UTAUT model, where social influence not only shapes attitude towards investment but can also directly translate into technology usage behavior [73], and in this case, investment behavior. In line with the above discussion, the study suggests the following research hypothesis.

Hypothesis H7: Social-media financial influencers have a positive influence on investment decisions.

A positive attitude toward investment does not necessarily translate into actual behavior unless individuals believe in their ability to act [56]. Fintech self-efficacy reflects an individual's confidence in using digital financial services [34]. This confidence is captured by fintech self-efficacy, which reflects an individual's capability to manage financial goals through digital services, including saving, borrowing, and using technological tools for investment [74]. FinTech platforms make cryptocurrency transactions more accessible and efficient. However, they also expose investors to higher volatility. The fast pace and interactive features of digital trading can cause investors to make impulsive decisions [75].

According to Self-Efficacy Theory, individuals with higher fintech self-efficacy feel more capable of executing investment actions, which reinforces the link between positive attitudes and actual investment behavior [37]. Empirical studies have shown that fintech self-efficacy strengthens the relationship between attitudes and investment behavior in contexts such as online banking [76], and general financial planning [77]. In cryptocurrency markets, this mechanism becomes even more salient because digital platforms reduce transactional friction, provide real-time market information, and enable rapid execution of trades. These streamlined processes shorten decision time and can unintentionally amplify risk-taking, especially in highly volatile environments. Individuals with stronger fintech self-efficacy may therefore act more quickly and decisively on their attitudes when engaging with digital trading interfaces. Examining this moderating role in the crypto context extends self-efficacy theory and clarifies how confidence in using FinTech shapes investment choices in emerging digital markets. Based on this, the study proposes the following research hypothesis.

Hypothesis H8: Fintech self-efficacy positively moderates the relationship between attitude toward investment and investment decision.

From the perspective of Behavioral Finance Theory, impulsivity is a behavioral bias that undermines rationality, while literacy acts as a buffer against such bias [39]. Although this pathway has not been directly tested in cryptocurrency markets, theoretical and indirect evidence support a mediating effect. Prior studies indicate that higher levels of digital financial literacy reduce impulsive behavior [45,48], which in turn improves the quality of investment decisions [78]. Investors with stronger knowledge are better able to anticipate long-term outcomes, control emotions, and resist impulsive actions [79,80]. In contrast, limited literacy often leads to hasty decisions driven by external cues [81]. Given the discussion, the following research hypothesis is formulated:

Hypothesis H9. Impulsivity plays a mediating role in the relationship between digital financial literacy and investment decisions.

Digital financial literacy also shapes investment behavior through its influence on attitudes. By improving the ability to evaluate risks and benefits, literacy fosters positive beliefs that translate into favorable attitudes toward investment [20,82]. The Theory of Planned Behavior highlights attitudes as a central mechanism that converts beliefs into actions [35]. Empirical findings further show that attitudes significantly mediate the impact of financial knowledge on investment outcomes [81].

Similarly, social media financial influencers shape attitudes that guide investment choices, especially among young investors [83,84]. This mechanism is consistent with UTAUT, which explains the role of social influence, while TPB emphasizes the mediating function of attitudes. Integrating these perspectives provides a comprehensive explanation of how social influence is converted into behavior. Accordingly, the study puts forward two hypotheses:

H10: Attitude towards investment plays a mediating role in the relationship between digital financial literacy and investment decisions.

H11: Attitude towards investment plays a mediating role in the relationship between social media financial influencers and investment decisions.

From the theoretical basis and research hypotheses, the article proposes a research model as shown in Fig 1. The model integrates technological, social, and behavioral perspectives to explain cryptocurrency investment behavior in Vietnam. Digital Financial Literacy represents a technological factor influencing Impulsivity, Attitude Toward Investment, and Investment Decision. Social-Media Financial Influencers capture the social factor shaping investor attitudes and behaviors, while Fintech Self-efficacy acts as a cognitive-behavioral factor moderating the link between attitude and decision. Impulsivity reflects behavioral tendencies, and Investment Decision serves as the behavioral outcome, representing actual investment behavior.

## 3. Research methodology

Data was collected through an online survey with 505 valid responses from individual investors participating in the cryptocurrency market. The data collection period is from June 1, 2025, to July 15, 2025. This sample size satisfies the reliability requirements of the PLS-SEM model, exceeding the minimum threshold recommended by Hair et al., and is large enough to ensure representativeness and statistical significance [85]. The sample was collected using a convenience method. It focused on forums, online communities, and discussion groups about cryptocurrencies. These are places where young investors from the

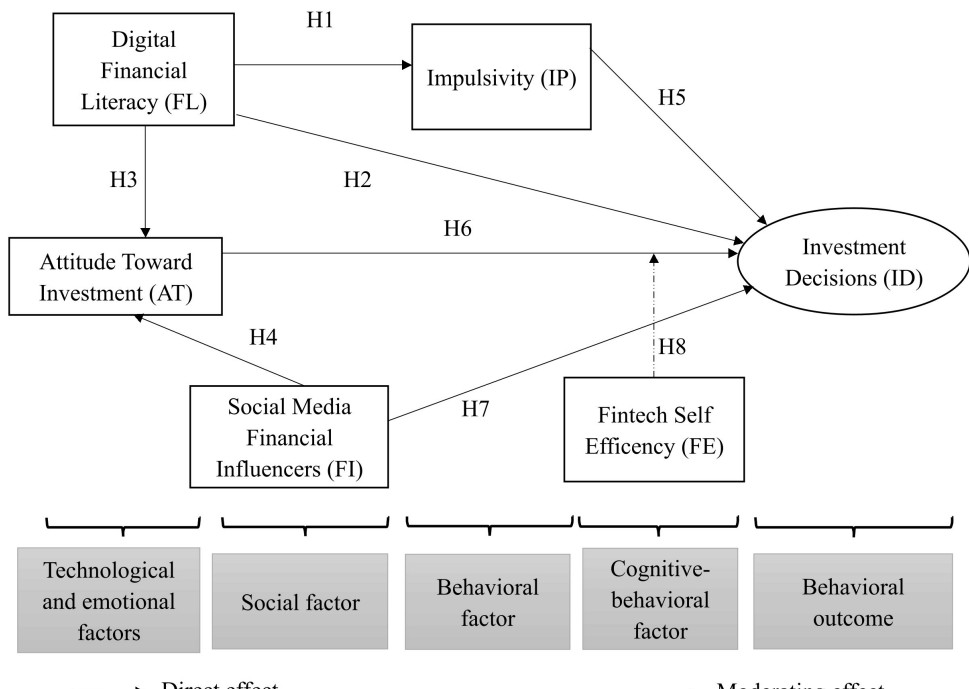

**Fig 1. Proposed research model.**

Millennial and Gen Z generations often gather. This group is highly engaged with financial technology. They regularly access information from social networks. Therefore, they are suitable for the research objective of studying investment behavior in the context of digitalization. The questionnaire was translated using the back-translation method to ensure semantic equivalence. A pilot test with 30 respondents was conducted to verify the reliability and clarity of items before full-scale data collection.

This study was approved by the Scientific Council of Ho Chi Minh University of Banking. All procedures involving human participants were conducted in accordance with institutional and national ethical standards. Before participation, all respondents were clearly informed about the research purpose, data confidentiality, and their right to withdraw at any stage. Written informed consent was obtained electronically before completing the questionnaire to ensure ethical compliance and voluntary participation.

The research variables were measured using a 7-point Likert scale, inherited and adjusted from previous studies to ensure reliability and conceptual validity. Digital Financial Literacy [57] reflects an individual's cognitive and practical ability to understand, analyze, and apply financial knowledge in digital and cryptocurrency investment contexts. Attitude Toward Investment [57] captures investors' positive or negative evaluations and beliefs regarding cryptocurrency as a viable investment option. Impulsivity represents the tendency of investors to make spontaneous, emotion-driven investment decisions without careful deliberation [43]. Investment Decision measures the behavioral intention and commitment to engage in cryptocurrency investment activities, including willingness to invest and recommend [86]. Social Media Financial Influencers [87] assess the extent to which investors rely on credible, expert, and trustworthy influencers for investment-related information and guidance. Finally, Fintech Self-Efficacy [74] reflects individuals' confidence in their ability to use financial technologies to effectively manage, plan, and achieve their financial goals. The use of scales that have been validated in various contexts allows for improved methodological rigor and ensures international comparability.

The data were analyzed using a partial least squares structural equation model (PLS-SEM). PLS-SEM was chosen due to the model's complexity, the presence of formative indicators, and the non-normal distribution of the data. PLS-SEM is also suitable for prediction-oriented and exploratory studies. The analysis process was carried out in two stages. First, the measurement model was assessed to check the reliability and validity of the scales. Then, the structural model was tested to examine the research hypotheses, including direct, mediating, and moderating effects. The bootstrapping technique with 5,000 replicates was applied to evaluate the statistical significance of the estimated coefficients. This approach ensures that the results are reliable and useful in explaining cryptocurrency investment behavior.

## 4. Research results

### 4.1. Statistical description

Table 2 presents the results of the statistical description of the research data. The survey included 505 participants. Most of them were Gen Z (under 25 years old) and Millennials (from 25–34 years old). These two groups made up 74.5% of the sample, showing the key role of young investors in the cryptocurrency market. In terms of gender, men predominated at 61.4%, while women accounted for 38.6%, indicating that the investment participation trend is still tilted towards men in the young age group. In terms of education level, the majority of participants had a university degree (57.2%) or a postgraduate degree (27.9%), demonstrating that young investors often have a solid academic background. Monthly income was mainly concentrated at 10–20 million VND (42.2%), suitable for the ability to pay for small and medium investments. Investment experience is mainly from less than 1 year to 3 years (67.9%), showing that many young investors are still in the learning and experience accumulation stage.

### 4.2. Measurement model evaluation results

From the measurement model evaluation results in Table 3, it is evident that all the scales in the study satisfy the requirements of reliability and convergent validity. This confirms their appropriateness for use in the subsequent analyses. First of all, in terms of reliability, all scales have Cronbach's Alpha and Rho_A values exceeding 0.80, reflecting high internal

**Table 2. Demographic statistics.**

| Demographic characteristics | Classification | Frequency (n) | Ratio (%) |
|---|---|---|---|
| **Gender** | Male | 310 | 61.4 |
| | Female | 195 | 38.6 |
| **Age** | < 25 | 162 | 32.1 |
| | 25–34 | 214 | 42.4 |
| | 35–44 | 89 | 17.6 |
| | > 45 | 40 | 7.9 |
| **Education** | College | 75 | 14.9 |
| | Bachelor | 289 | 57.2 |
| | Postgraduate | 141 | 27.9 |
| **Income** | Under 10 million VND | 102 | 20.2 |
| | 10–20 million VND | 213 | 42.2 |
| | 20–30 million VND | 129 | 25.5 |
| | Over 30 million VND | 61 | 12.1 |
| **Investment experience** | Less than 1 year | 145 | 28.7 |
| | 1–3 years | 198 | 39.2 |
| | 3–5 years | 102 | 20.2 |
| | Over 5 years | 60 | 11.9 |

consistency between observed variables. This shows that the items in the same scale measure the same concept stably and reliably. This is an important indicator, because the stability and reliability of the scale are prerequisites for accurately explaining the relationships in the research model.

In terms of convergent validity, the outer loading results of the observed variables are all greater than 0.70, ranging from 0.725 to 0.908. This threshold shows that each observed variable has a high level of contribution to the latent concept it represents. At the same time, the AVE (Average Variance Extracted) values of the scales all exceed the requirement of 0.50, ranging from 0.635 to 0.747. This shows that most of the variance of the observed variables is explained by the latent concepts, instead of by measurement errors, thereby ensuring the convergence of the scale. The results demonstrate that the scale in the study meets the standards of reliability and convergent validity.

### 4.3. HTMT evaluation results

Discriminant validity was assessed using the Heterotrait–Monotrait Ratio (HTMT) as recommended by Henseler et al. [89]. All HTMT values among the constructs were found to be below the threshold of 0.85, indicating satisfactory discriminant validity. This confirms that each construct measures a distinct concept and there is no significant overlap among latent variables. Therefore, the measurement model demonstrates adequate discriminant validity (see Table 4).

Common method bias (CMB) was assessed using Harman's single-factor test, with the first factor explaining less than 40% of the total variance, indicating no serious bias. Additionally, inner VIF values were below 5.0 (see Table 3), confirming that multicollinearity and CMB are not major concerns [90]. Multivariate normality was tested [91]. The results indicated non-normal data (Mardia's multivariate skewness and kurtosis $p < 0.05$), confirming the suitability of PLS-SEM.

### 4.4. Structural model evaluation results

The structural model analysis results in Table 5 indicate that all research hypotheses are supported at a significance level of $p < 0.05$, demonstrating that the model fits well with the data collected from the sample. The

**Table 3. Measurement model evaluation results.**

| Symbol | Item | Outer loading | Cronbach's Alpha | Rho_A | EVA | VIF |
|---|---|---|---|---|---|---|
| **Digital Financial Literacy (FL) [57]** | | | | | | |
| FL1 | I have a good understanding of how digital financial platforms work for cryptocurrency investment. | 0.834 | 0.866 | 0.874 | 0.712 | 2.521 |
| FL2 | I am familiar with the digital tools and applications for cryptocurrency investment. | 0.830 | | | | 2.788 |
| FL3 | I have a clear understanding of current trends and risks in cryptocurrency markets. | 0.837 | | | | 2.950 |
| FL4 | I can use my knowledge to evaluate the legitimacy, risks, and security features of cryptocurrency platforms before making investment decisions. | 0.875 | | | | 2.790 |
| **Impulsivity (IP) [43]** | | | | | | |
| IP1 | I often invest in Cryptocurrency on impulse. | 0.830 | 0.830 | 0.849 | 0.664 | 1.792 |
| IP2 | I often invest in Cryptocurrency on the spur of the moment. | 0.842 | | | | 2.062 |
| IP3 | I often invest in Cryptocurrency without any deliberation. | 0.874 | | | | 2.221 |
| IP4 | I generally invest in Cryptocurrency without thinking too much. | 0.703 | | | | 1.453 |
| **Investment decisions (ID) [88]** | | | | | | |
| ID1 | I plan to invest in cryptocurrency. | 0.848 | 0.832 | 0.870 | 0.663 | 2.093 |
| ID2 | I strongly recommend others to invest in cryptocurrency. | 0.902 | | | | 2.678 |
| ID3 | I will continue to invest in cryptocurrency. | 0.761 | | | | 1.837 |
| ID4 | I can stand the inconvenience caused by cryptocurrency investment. | 0.734 | | | | 1.623 |
| **Attitude Toward Investment (AT) [57]** | | | | | | |
| AT1 | It's a good idea to invest money in the cryptocurrency market. | 0.861 | 0.865 | 0.867 | 0.788 | 1.985 |
| AT2 | The cryptocurrency market is a sensible investment option. | 0.903 | | | | 2.471 |
| AT3 | I like the idea of investing money in the cryptocurrency market. | 0.898 | | | | 2.428 |
| **Social media financial influencers (FI) [87]** | | | | | | |
| FI1 | I follow and take investment advice from expert financial influencers. | 0.903 | 0.877 | 0.885 | 0.732 | 3.471 |
| FI2 | I follow financial influencers who provide up-to-date and relevant crypto-currency information. | 0.901 | | | | 3.489 |
| FI3 | I follow and rely on trusted financial influencers for cryptocurrency invest-ment decisions. | 0.816 | | | | 1.738 |
| FI4 | I follow and take investment advice from reliable financial influencers. | 0.796 | | | | 1.860 |
| **Fintech self-efficacy (FE) [74]** | | | | | | |
| FE1 | I am confident that I can manage my finances. | 0.903 | 0.890 | 0.922 | 0.695 | 4.037 |
| FE2 | I can manage to spend less than I earn every month. | 0.837 | | | | 3.273 |
| FE3 | I can deposit money in the bank with confidence to prepare for the future. | 0.720 | | | | 1.849 |
| FE4 | I can borrow money from the bank. | 0.886 | | | | 3.970 |
| FE5 | I can use financial services to manage my finances | 0.811 | | | | 2.787 |

bootstrapping results indicate that all estimated paths are statistically significant at the 95% confidence level, as none of the confidence intervals include zero, confirming the robustness and reliability of the structural relationships.

Among the predictors, Attitude Toward Investment (H6, β = 0.381, p < 0.001) and Digital Financial Literacy (H2, β = 0.246, p < 0.001) exert the strongest direct influences on Investment Decision, followed by Social Media Financial Influencers (H7, β = 0.220, p < 0.001) and Impulsivity (H5, β = 0.163, p < 0.05). This indicates that both cognitive factors and social influence play important roles in shaping cryptocurrency investment decisions.

**Table 4. HTMT evaluation results.**

|        | AT    | FE    | FI    | FL    | ID    | IP    |
|--------|-------|-------|-------|-------|-------|-------|
| FE     | 0.571 |       |       |       |       |       |
| FI     | 0.790 | 0.495 |       |       |       |       |
| FL     | 0.736 | 0.587 | 0.737 |       |       |       |
| ID     | 0.739 | 0.535 | 0.591 | 0.627 |       |       |
| IP     | 0.825 | 0.689 | 0.757 | 0.839 | 0.697 |       |
| FE x AT| 0.402 | 0.451 | 0.242 | 0.330 | 0.377 | 0.344 |

**Table 5. Structural model evaluation results.**

| Hypothesis | Relationship | Path coefficient | t-value | p-value | [2.5%, 97.5%] |
|------------|--------------|------------------|---------|---------|---------------|
| H1 | Digital Financial Literacy→Impulsivity | 0.726 | 31.847 | 0.000 | [0.680, 0.769] |
| H2 | Digital Financial literacy→Investment decision | 0.246 | 4.514 | 0.000 | [0.142, 0.352] |
| H3 | Digital Financial Literacy→Attitude Toward Investment | 0.335 | 7.395 | 0.000 | [0.247, 0.426] |
| H4 | Social media financial influencers→Attitude Toward Investment | 0.477 | 11.032 | 0.000 | [0.387, 0.557] |
| H5 | Impulsivity →Investment decision | 0.163 | 2.212 | 0.027 | [0.017, 0.306] |
| H6 | Attitude Toward Investment→Investment decision | 0.381 | 5.994 | 0.000 | [0.258, 0.510] |
| H7 | Social media financial influencers→Investment decision | 0.220 | 4.477 | 0.000 | [0.126, 0.318] |
| H8 | Fintech self-efficacy x Attitude Toward Investment→Investment decision | −0.082 | 2.140 | 0.032 | [-0.154, -0.001] |
| H9 | Digital Financial literacy→Impulsivity→Investment decision | 0.118 | 2.215 | 0.027 | [0.012, 0.221] |
| H10 | Digital Financial literacy→Attitude Toward Investment→Investment decision | 0.128 | 4.956 | 0.000 | [0.081, 0.182] |
| H11 | Social media financial influencers→Attitude Toward Investment→Investment decision | 0.182 | 4.959 | 0.000 | [0.115, 0.259] |

The interaction effect between fintech self-efficacy and attitude toward investment is negative and statistically significant (H8, $\beta = -0.082$, $p < 0.05$), indicating that fintech self-efficacy weakens, rather than strengthens, the impact of investment attitudes on actual investment decisions. In practical terms, this means that when investors are more confident in using financial technologies, positive attitudes toward investment are less likely to directly translate into action. Instead of relying on their general attitudes, technologically confident investors appear to make decisions more independently, based on their own judgment and operational capability. Although the size of this moderating effect is relatively small, such effects are common in interaction terms and remain relevant in digital investment settings, where even modest differences in perceived technological competence can alter how investors act on their attitudes.

Specifically, Digital Financial Literacy has a strong positive impact on Impulsivity (H1, $\beta = 0.726$, $p < 0.001$), indicating that higher digital financial literacy is associated with greater confidence and a higher tendency toward quick decision-making. This positive direction suggests that digital financial literacy in the Vietnamese context may enhance investors' engagement and responsiveness to market information rather than purely restraining impulsive behavior. Although Although Hypothesis H1 originally proposed a negative relationship, the empirical results show a strong positive association between digital financial literacy and impulsivity. This indicates that, in cryptocurrency markets, higher digital financial literacy does not necessarily constrain impulsive behavior but may instead increase confidence and perceived control, thereby amplifying affect-driven investment actions [92]. In this study, impulsivity is measured through spontaneous and minimally deliberated investment behaviors, capturing emotional rashness rather than informed decisiveness [43]. Accordingly, the positive relationship should not be interpreted as rationally faster decision-making. Rather, it reflects

overconfidence and illusion of control biases, whereby digitally literate investors, often influenced by informal online information and social media, overestimate their ability to interpret market signals in highly volatile environments.

Digital financial literacy also exerts a direct and meaningful positive effect on Investment Decision (H2, β = 0.336, p < 0.001) and a significant positive influence on Attitude Toward Investment (H3, β = 0.335, p < 0.001). These results imply that financial literacy strengthens investors' positive attitudes and confidence in making investment decisions, consistent with previous studies [1,93]. This aligns with Self-Efficacy Theory, which suggests that financial knowledge and skills enhance behavioral control and strengthen beliefs about one's ability to invest effectively [20,94]. However, in Vietnam, digital financial literacy often develops through informal learning, such as social media or online communities, rather than formal education. This partial knowledge may increase investors' confidence but could also lead to overconfidence, reinforcing some degree of impulsivity in volatile cryptocurrency markets.

In addition, Social Media Financial Influencers significantly affect Attitude Toward Investment (H4, β = 0.477, p < 0.001) and also exert a direct and statistically significant influence on Investment Decision (H7, β = 0.220, p < 0.001). This highlights that the influence of financial influencers primarily operates through shaping attitudes rather than directly altering investment behavior. This finding underscores the importance of social networks in forming perceptions and attitudes of young investors, particularly Gen Z and Millennials. These investors are strongly influenced by information from KOLs, online communities, and reference groups, supporting the argument that personal financial behavior is shaped by social norms [59]. In Vietnam's collectivist culture, where group belonging and community trust are emphasized, such social influence is amplified, meaning attitude formation becomes a critical mediating mechanism between social exposure and investment behavior.

Investment attitudes further play a critical mediating role in the model. Attitude Toward Investment positively influences Investment Decision (H6, β = 0.381, p < 0.001) and mediates the effects of Digital Financial Literacy (H10, β = 0.128, p < 0.001) and Social Media Financial Influencers (H11, β = 0.182, p < 0.001) on Investment Decision. Moreover, Impulsivity also has a significant positive effect on Investment Decision (H5, β = 0.163, p < 0.05) and partially mediates the relationship between Digital Financial Literacy and Investment Decision (H9, β = 0.118, p < 0.05). These mediation results confirm that the model captures not only direct effects but also the indirect behavioral mechanisms through which cognitive and social factors influence investment behavior.

Beyond cryptocurrency, investor behavior in Vietnam follows patterns commonly seen in Asian financial markets, where retail investors dominate trading activity and influence market movements. Studies on Asian markets show that individual investors tend to trade more actively during periods of uncertainty, often adopting contrarian or opportunity-seeking strategies, while foreign institutional investors usually reduce exposure or exit local markets during market shocks [95]. Placing the Vietnamese cryptocurrency market within this regional context helps explain the observed behaviors and strengthens the generalizability of the study's findings.

These results are consistent with prior studies [81], showing that digital financial literacy and social influence, mediated through attitudes and impulsivity, play central roles in shaping investment behavior. They emphasize that young investors often combine rational factors (knowledge, self-efficacy) with emotional and social factors (influence, impulsivity) when making investment decisions. This provides a nuanced cultural insight into how Vietnamese investors integrate analytical reasoning, emotional engagement, and social validation in cryptocurrency investment behavior.

The analysis results in Table 6 show that the factors explaining investment decision (ID) reached $R^2 = 0.488$, reflecting a strong average influence, while attitude toward investment (AT) and Impulsivity (IP) also achieved significant $R^2$ values (0.545 and 0.526), indicating that the model has good explanatory power.

Regarding $f^2$, digital financial literacy (FL) had a small effect on investment decisions ($f^2 = 0.006$). At the same time, it exerted a large influence on impulsivity ($f^2 = 1.111$) and a small effect on attitude toward investment ($f^2 = 0.143$). This shows that digital financial literacy plays a particularly important role in shaping impulsive tendencies, while its direct contribution to investment decisions remains limited.

**Table 6. Results of f² and Q².**

| Path | f² Value | Effect Size Interpretation | R² | Q² | Predictive Relevance |
|---|---|---|---|---|---|
| **AT → ID** | 0.109 | Small | 0.488 | 0.304 | Medium |
| **FI→ID** | 0.095 | Small | | | |
| **FL→ID** | 0.006 | Small | | | |
| **IP→ID** | 0.018 | Small | | | |
| **FI → AT** | 0.291 | Medium | 0.545 | 0.425 | Large |
| **FL → AT** | 0.143 | Medium | | | |
| **FL→IP** | 1.111 | Large | 0.526 | 0.345 | Medium |
| **FE × AT → ID** | 0.007 | Small | | | |

Social media financial influencers (FI) had a medium effect on attitude toward investment (f²=0.291) but only a negligible effect on investment decisions (f²=0.095). Fintech self-efficacy (FE) demonstrated a small moderate effect on the relationship between investment decisions and attitude toward investment (FE × AT → ID; f²=0.007). Additionally, attitude toward investment had a small effect on investment decisions (f²=0.109), while impulsivity showed a negligible effect (f²=0.018).

The Q² values are all positive, confirming that the model has predictive relevance in the research context. Specifically, Investment Decision (Q²=0.304) exhibits medium predictive relevance, Attitude Toward Investment (Q²=0.425) shows large predictive relevance, and Impulsivity (Q²=0.345) demonstrates medium predictive relevance according to the thresholds suggested by Hair et al. [96]. These results indicate that the structural model has a strong predictive capacity for explaining behavioral outcomes, particularly in explaining attitudes toward investment.

## 5. Conclusions and policy implications

### 5.1. Conclusions

This study confirms that digital financial literacy plays a central role in shaping cryptocurrency investment behavior in Vietnam. Rather than reducing impulsivity, higher digital financial literacy is associated with greater confidence, faster market responsiveness, and more active decision-making, even when such decisions are partly affect-driven. Social media financial influencers influence investment behavior both directly and indirectly through investment attitudes, while fintech self-efficacy moderates the relationship between attitude and investment decision by reducing reliance on attitudinal cues. These findings highlight that Vietnamese investors' decisions are influenced by a combination of cognitive, social, and technological factors. Analytical reasoning coexists with emotional engagement and community influence in the dynamic context of the cryptocurrency market.

### 5.2. Theoretical contributions

This study advances behavioral finance theory by clarifying the multifaceted role of digital financial literacy in cryptocurrency investment behavior. Rather than merely mitigating behavioral biases, digital financial literacy enhances investors' confidence, analytical capability, and responsiveness to digital market dynamics. The findings demonstrate that literacy may simultaneously promote more structured attitudes and stronger impulsive tendencies, reflecting a dual cognitive–behavioral mechanism in high-volatility environments.

The findings extend the TPB by confirming the mediating role of Attitude Toward Investment, which translates cognitive (digital literacy) and social (influencer exposure) determinants into behavioral outcomes. They also extend the UTAUT and

Use of Technology by demonstrating that social influence from financial influencers indirectly affects investment behavior through both attitudinal mechanisms and direct behavioral pathways, emphasizing the role of perceived norms in online financial communities.

Furthermore, the results contribute to Self-Efficacy Theory by showing that fintech self-efficacy acts as a moderator, shaping how attitudes translate into investment actions by slightly attenuating the strength of the attitude–decision relationship. This negative moderating effect suggests that individuals with higher fintech self-efficacy may rely less exclusively on attitudinal cues and more on autonomous technological competence when making decisions. Empirically, the integration of UTAUT's social influence construct with TPB's attitudinal process provides evidence of theoretical convergence between technology adoption and behavioral intention models, highlighting the combined influence of cognitive, social, and technological factors in shaping cryptocurrency investment behavior.

## 5.3. Practical contributions

In light of the finding that higher digital financial literacy is positively associated with impulsivity in cryptocurrency investment, policy implications should move beyond a purely technical focus. Programs that emphasize digital or technical financial skills alone may be insufficient and could unintentionally amplify impulsive trading behavior in highly volatile markets. The empirical results indicate that higher perceived digital competence is associated with greater confidence and perceived control, and these characteristics are linked to more spontaneous and affect-driven investment actions in fast-moving cryptocurrency markets. Therefore, investor education initiatives should integrate behavioral discipline, emotional regulation, and self-control alongside technical knowledge to promote more sustainable investment behavior.

In terms of policy implications, regulators and financial institutions may prioritize behaviorally informed investor education programs that address both cognitive and social aspects of decision making. Evidence based initiatives involving universities, fintech firms, and government agencies can help promote digital literacy and responsible financial engagement among high-risk groups, particularly young and first-time investors. Given that social media financial influencers are positively associated with investment decisions both directly and indirectly through attitudes, regulatory frameworks may consider enhancing transparency and disclosure standards, as well as strengthening accountability mechanisms for online financial content. Rather than pursuing broad regulatory reforms, the focus should be on feasible, high-impact measures that strengthen transparency, enhance digital competence, and encourage more critical engagement with social media–based financial information in investment decisions.

## 5.4. Policy implications

The findings provide several important implications for policymakers, regulators, and market participants. In light of the result that higher digital financial literacy may be associated with greater impulsivity in cryptocurrency investment, policy interventions should move beyond a purely technical approach. While digital financial education remains important, programs that focus solely on expanding analytical or technical skills may be insufficient and could unintentionally intensify impulsive trading in highly volatile markets. Therefore, digital financial education initiatives should explicitly integrate behavioral discipline, emotional regulation, and self-control alongside technical knowledge to support more sustainable investment behavior.

Vietnam should continue to develop digital financial education programs tailored to the needs of young investors. These programs should be voluntary, respect user privacy, and focus on building basic analytical and risk-management skills together with behavioral awareness. This approach supports informed and independent decision-making.

Vietnam also needs a transparent regulatory framework to oversee financial influencers. Social media financial influencers affect investment decisions both directly and through attitudes. Regulators should therefore require clear disclosure, transparent sponsorship information, and stronger accountability for online financial content. This framework should

clearly define the scope of oversight and avoid excessive monitoring. Its main purpose is to ensure that financial information shared online is accurate and trustworthy, without restricting legitimate discussion.

In higher education, universities may incorporate trading simulations and decision-making courses into economics and technology programs. These modules should be flexible and optional, allowing students to strengthen independent thinking while maintaining freedom of choice.

For banks and fintech companies, application design should emphasize transparency. Risk-checking tools, expense tracking, and personalized suggestions should be opt-in features, enabling users to control how their data is used. This helps reduce algorithmic bias and protects user autonomy. Because fintech self-efficacy shapes how attitudes turn into actual investment actions, platforms should promote informed decision-making instead of simply enabling faster transactions.

Public communication campaigns also play an important role. These campaigns should present both the benefits and risks of cryptocurrency, helping investors understand that crypto is not a shortcut to quick wealth but a high-risk asset that requires careful preparation.

These measures contribute to a safer and more transparent digital financial environment. Balancing technological competence with behavioral discipline and emotional regulation is essential for the long-term and sustainable development of Vietnam's digital financial market.

## 6. Limitations of the study

This study is not without limitations. First, the use of self-reported online surveys may introduce cognitive and social desirability biases, despite efforts to minimize common method variance. Future studies could mitigate this limitation by combining self-reports with behavioral or transactional data from financial platforms to enhance data reliability. Second, the sample is limited to young Vietnamese investors, restricting the generalizability of the findings to other age groups, professional investors, or different market contexts. Subsequent research could expand to more diverse demographic segments or include comparative samples from different regions to capture variations in financial behavior. Third, the cross-sectional design only captures relationships at one point in time, which prevents causal inferences. Applying longitudinal or panel data approaches could help validate causal relationships and track the evolution of investment behavior over time. Future research could address these issues by adopting longitudinal designs to track changes over time, employing mixed-method approaches to enrich insights with qualitative evidence, and extending the scope to cross-country comparisons. Additionally, future studies may explore new moderating variables such as perceived financial regulation, risk awareness, or trust in cryptocurrency platforms to deepen theoretical understanding. Such efforts would strengthen external validity and reveal cultural or institutional differences that influence the role of financial literacy, social influence, and fintech self-efficacy in shaping cryptocurrency investment behavior. Future research may distinguish objective digital financial literacy from perceived literacy to better explain their differential effects on impulsive investment behavior.

## Supporting information

**S1 File. Output data files.** These data include the full analytical outputs used to generate the results reported in the manuscript.
(ZIP)

## Acknowledgments

During the preparation of this manuscript, the author used ChatGPT for grammar improvement only, including minor corrections to punctuation and language fluency. After using this tool, the author thoroughly reviewed and edited all content to ensure accuracy and originality, and takes full responsibility for the final version of the publication.

## Author contributions

**Conceptualization:** Tang My Sang.

**Data curation:** Tang My Sang.

**Formal analysis:** Tang My Sang.

**Funding acquisition:** Tang My Sang.

**Investigation:** Tang My Sang.

**Methodology:** Tang My Sang.

**Project administration:** Tang My Sang.

**Resources:** Tang My Sang.

**Software:** Tang My Sang.

**Supervision:** Tang My Sang.

**Validation:** Tang My Sang.

**Visualization:** Tang My Sang.

**Writing – original draft:** Tang My Sang.

**Writing – review & editing:** Tang My Sang.

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
