## [Decision Letter · Decision Letter 0]

27 Oct 2025

Dear Dr. Tang,

Thank you for submitting your manuscript to PLOS ONE. After careful consideration, we feel that it has merit but does not fully meet PLOS ONE’s publication criteria as it currently stands. Therefore, we invite you to submit a revised version of the manuscript that addresses the points raised during the review process.

We look forward to receiving your revised manuscript.

Kind regards,

Nishi Malhotra

Academic Editor

PLOS ONE

[This research is funded by Ho Chi Minh University of Banking, Ho Chi Minh city, Vietnam].

3. In the online submission form, you indicated that [The data supporting this study's findings are available on request from the author. The data are not publicly available due to the privacy of research participants].

Reviewers' comments:

Reviewer's Responses to Questions

**Comments to the Author**

1. Is the manuscript technically sound, and do the data support the conclusions?

Reviewer #1: Yes

Reviewer #2: Partly

2. Has the statistical analysis been performed appropriately and rigorously?

Reviewer #1: Yes

Reviewer #2: Yes

3. Have the authors made all data underlying the findings in their manuscript fully available?

Reviewer #1: Yes

Reviewer #2: No

4. Is the manuscript presented in an intelligible fashion and written in standard English?

Reviewer #1: Yes

Reviewer #2: Yes

Reviewer #1: The authors employ a Structural Equation Model (SEM) to analyze the investment decision-making process of young investors in cryptocurrency markets in Vietnam. The question is relevant and interesting, and the sample used appears to be particularly noteworthy. However, three main issues should be addressed to make the analysis publication-worthy:

1. Meaning of Variables

First and most importantly, authors must clarify what their primary outcome variable means: Investment Decisions (ID). Prima facie, one would think that it refers to the decision to participate or not in cryptocurrency markets, which is what the research hypothesis development seems to imply. However, once we examine the actual questions that comprise this construct and carefully read Juwitta et al. (2022), where the questions originated, we realize that it may be risk aversion that the authors are actually measuring. This outcome variable must be made more explicit; otherwise, the entire SEM is completely misspecified.

2. Conceptual Map

Even if the ID variable were defined correctly, the article lacks a clear conceptual map that justifies the need to test the influence of all the variables proposed in ID. The authors employ a combination of UTAUT, SET, and TPB in their measurement equation, but fail to justify why the specific mix of constructs proposed should determine ID. They just list how different theories may explain the behavior. More is needed. The authors should at least discuss how the theories used may complement each other and discuss why they should be combined.

Finally, the discussion of Impulsivity is not clear enough, either, but this is probably a result of the indeterminacy surrounding the ID variable.

3. Measurement of Variables

The definitions and names of the variables are unclear. I understand that the authors use previously published variables; however, the naming conventions and wording of the questions are not recognizable to an international audience. For example, the variable named "Investment decisions" includes questions like "You will never hang-gliding or bungee jumping in the cryptocurrency market". This wording may be influenced by Vietnamese conventions or Gen Z speech patterns, but it lacks context and therefore makes little sense to a broader audience.

This is also a problem with measuring Digital Financial Literacy. Question F4 hardly measures financial literacy. It measures the investor's confidence or overconfidence.

The attitude towards investment variable also has problems, as question AT3 refers to stocks, not cryptocurrency.

4. Minor issues.

- The discussion of behavioral finance is too vague and seems to equate it with the study of Impulsivity. This is not the case.

- The claim that research has "seldom integrated psychological, social, and technological factors into a unified model" is far too strong.

- The argument that FinTech facilitates transactions but also increases risk, given its importance to the hypothesis development, should be made explicit, not just dealt with in one reference.

Reviewer #2: The manuscript addresses a relevant and contemporary topic by integrating behavioral, social, and technological determinants of cryptocurrency investment. However, despite its comprehensive coverage and sound structure, several methodological, theoretical, and presentation weaknesses reduce its rigor and originality. While the topic aligns with PLOS ONE’s interdisciplinary scope, the paper requires substantial revision before it can be considered for publication.

1. Title, Abstract, and Keywords

• The title is clear but overly long. It may be shortened to emphasize the central contribution (e.g., “Integrating Behavioral, Social, and Technological Factors in Cryptocurrency Investment Decisions”).

• The claim that “research has seldom integrated psychological, social, and technological factors” is overstated—recent studies (2023–2025) have done so, including those cited later (e.g., Rijanto & Utami, 2024; Hassan et al., 2024).

2. Introduction

• The introduction is detailed but too long and descriptive, reading more like a literature review than a problem statement.

• The research gap is stated in general terms (lack of integrated frameworks) but not theoretically or empirically substantiated. The authors should demonstrate how their model advances behavioral finance theory beyond existing TPB–UTAUT integrations already seen in recent work (e.g., Bouteraa, 2024; Hassan et al., 2024).

• The motivation for focusing on Vietnam is clear but underdeveloped; comparative insights or justification for cultural relevance (e.g., collectivist society, crypto regulation) would strengthen the contextual contribution.

3. Theoretical Framework and Hypotheses

• The integration of four theories (UTAUT, TPB, Self-efficacy, Behavioral Finance) is ambitious but insufficiently justified. The paper lists theories rather than explaining how their constructs conceptually interlock.

• Some hypotheses are mechanical restatements of prior literature rather than theory-driven. For example:

o H5 (Impulsivity → Investment Decision) lacks novelty and empirical justification for cryptocurrency specifically.

o H8 (moderation by Fintech Self-Efficacy) should include theoretical rationale—why does self-efficacy strengthen, not weaken, the attitude–behavior link?

• Several hypotheses (H9–H11) use mediation/moderation terms interchangeably without explicit conceptual diagrams or theoretical logic for directionality.

4. Methodology

• Measurement scales are adopted from diverse sources, but no evidence of translation process, or pilot testing is provided. Reliability is reported (Cronbach’s alpha), yet discriminant validity is only briefly assessed (HTMT table) but the discussion seems as Pearson correlation discussion.

• The study uses PLS-SEM, which is suitable for exploratory models; however, justification for choosing PLS over covariance-based SEM is not provided. The authors should explain why PLS was used (e.g., non-normal data, model complexity).

• Common method bias (CMB) is likely, as all data are self-reported. The paper should apply Harman’s single-factor test or VIF statistics to mitigate this concern.

• Multivariate normality is not assessed.

5. Results and Interpretation

• All eleven hypotheses are supported, which is statistically improbable and suggests potential model overfitting or confirmation bias.

• confidence intervals should be reported

• The f² and Q² table is mislabelled (“Bảng” indicates incomplete editing). Effect sizes are inconsistently interpreted—particularly for the mediated relationship, are described without backing.

• The mediation results (H9–H11) are not shown through indirect path coefficients or confidence intervals; these are essential for transparency.

• The interpretation remains descriptive, repeating numerical findings without deeper behavioral or cultural interpretation (e.g., why impulsivity still exerts a positive effect in Vietnam).

6. Discussion and Contributions

• The discussion section repeats the results rather than linking them to broader behavioral finance or fintech literature.

• The claimed theoretical contributions (extending TPB, UTAUT, self-efficacy) are not demonstrated empirically; instead, they remain declarative.

• Practical implications are extensive but lack prioritization or feasibility analysis (e.g., “regulatory oversight” and “public education campaigns” are policy-level actions requiring evidence-based design).

7. Limitations and Future Research

• The limitations are appropriately listed (self-report, cross-sectional, young sample), but no mitigation strategies are suggested. Including remedies (e.g., longitudinal or multi-source data) would enhance credibility.

• Future research directions should identify specific constructs or moderating variables (e.g., financial regulation awareness, perceived risk, or crypto trust) rather than restating standard methodological fixes.

8. Writing Quality and Formatting

• The manuscript is mostly well-written but shows signs of AI-assisted phrasing—overly polished transitions (“Taken together,” “This study enriches,” etc.) and uniform sentence rhythm.

• Some references lack full details or formatting consistency.

• Figures and tables are not numbered or cross-referenced properly (e.g., “Fig 1” lacks caption clarity).

9. Ethical and Data Transparency

• The data availability statement (“available on request”) conflicts with PLOS ONE’s open-data policy. Data should be deposited in a public repository with anonymized files unless privacy law prohibits it.

• The AI acknowledgment is transparent but should include the exact purpose (“used for grammar improvement only”).

**Do you want your identity to be public for this peer review?** For information about this choice, including consent withdrawal, please see our For information about this choice, including consent withdrawal, please see our Privacy Policy .

Reviewer #1: No

Reviewer #2: No

---

## [Author Response · Author response to Decision Letter 1]

13 Nov 2025

I would like to express my sincere gratitude to the reviewers for their insightful and constructive comments. In the revised manuscript, all modifications made in response to the reviewers’ suggestions.

Review #1

The authors employ a Structural Equation Model (SEM) to analyze the investment decision-making process of young investors in cryptocurrency markets in Vietnam. The question is relevant and interesting, and the sample used appears to be particularly noteworthy. However, three main issues should be addressed to make the analysis publication-worthy:

1. Meaning of Variables

First and most importantly, authors must clarify what their primary outcome variable means: Investment Decisions (ID). Prima facie, one would think that it refers to the decision to participate or not in cryptocurrency markets, which is what the research hypothesis development seems to imply. However, once we examine the actual questions that comprise this construct and carefully read Juwitta et al. (2022), where the questions originated, we realize that it may be risk aversion that the authors are actually measuring. This outcome variable must be made more explicit; otherwise, the entire SEM is completely misspecified.

Responses:

Thanks for the helpful feedback. I have clarified the definition and meaning of the variable “Investment Decisions”. This variable reflects the level of willingness and commitment to invest in the cryptocurrency market, not “risk aversion”. The scales have been recalibrated:

- “I plan to invest in cryptocurrency”,

- “I will continue to invest in cryptocurrency”,

- “I strongly recommend others to invest in cryptocurrency”

- “I can stand the inconvenience caused by cryptocurrency investment”.

I have also edited the “Measurement Scales” section for consistency.

2. Conceptual Map

Even if the ID variable were defined correctly, the article lacks a clear conceptual map that justifies the need to test the influence of all the variables proposed in ID. The authors employ a combination of UTAUT, SET, and TPB in their measurement equation, but fail to justify why the specific mix of constructs proposed should determine ID. They just list how different theories may explain the behavior. More is needed. The authors should at least discuss how the theories used may complement each other and discuss why they should be combined.

Finally, the discussion of Impulsivity is not clear enough, either, but this is probably a result of the indeterminacy surrounding the ID variable.

Responses:

I would like to thank the reviewer for his insightful comments.

In the revised version, I added a synthesis that shows the relationship between the four foundational theories including UTAUT, TPB, Self-efficacy Theory and Behavioral Finance Theory. Specifically, UTAUT explains the technological factor and social influence; TPB reflects the cognitive and intentional behavior aspects; Self-efficacy moderates the relationship between investment attitudes and behavior; while Behavioral Finance adds the emotional and impulsive factors.

In Table 1. Summary of underpinning theories, I added the Limitation of each theory in Cryptocurrency context to clarify the reason for the combination of theories

In addition, I also show the basis of combining the ideas integrated in the model, which includes the rational, social, personal ability and irrational aspects in investment decisions.

3. Measurement of Variables

The definitions and names of the variables are unclear. I understand that the authors use previously published variables; however, the naming conventions and wording of the questions are not recognizable to an international audience. For example, the variable named "Investment decisions" includes questions like "You will never hang-gliding or bungee jumping in the cryptocurrency market". This wording may be influenced by Vietnamese conventions or Gen Z speech patterns, but it lacks context and therefore makes little sense to a broader audience.

This is also a problem with measuring Digital Financial Literacy. Question F4 hardly measures financial literacy. It measures the investor's confidence or overconfidence.

The attitude towards investment variable also has problems, as question AT3 refers to stocks, not cryptocurrency.

Responses:

Thank you for your valuable comments.

In the revised version, I clarified the definition of each research variable (in the Research methodology section), and recalibrated the scale of Digital Financial Literacy, Investment decisions, Attitude Toward Investment (AT3), and Social-media financial influencers.

4. Minor issues.

- The discussion of behavioral finance is too vague and seems to equate it with the study of Impulsivity. This is not the case.

- The claim that research has "seldom integrated psychological, social, and technological factors into a unified model" is far too strong.

- The argument that FinTech facilitates transactions but also increases risk, given its importance to the hypothesis development, should be made explicit, not just dealt with in one reference.

Responses:

Thanks for your feedback.

- I have expanded and clarified the discussion of Behavioral Finance Theory, emphasizing that it is a general theory of cognitive and psychological biases that influence financial decisions, with Impulsivity being considered as just one specific behavioral bias.

- In the revised version, the sentence "seldom integrated psychological, social, and technological factors into a unified model" has been adjusted to read: “Although several studies have examined psychological, social, and technological factors separately, few have comprehensively integrated them into a unified framework.”

- Add the dual role of FinTech.

Review #2

1. Title, Abstract, and Keywords

The title is clear but overly long. It may be shortened to emphasize the central contribution (e.g., “Integrating Behavioral, Social, and Technological Factors in Cryptocurrency Investment Decisions”).

The claim that “research has seldom integrated psychological, social, and technological factors” is overstated—recent studies (2023–2025) have done so, including those cited later (e.g., Rijanto & Utami, 2024; Hassan et al., 2024).

Responses:

Thank you very much for your guidance, the title was revised as “Integrating Behavioral, Social, and Technological Factors in Cryptocurrency Investment Decisions”.

In the revised version, the sentence "seldom integrated psychological, social, and technological factors into a unified model" has been adjusted to read: “Although several studies have examined psychological, social, and technological factors separately, few have comprehensively integrated them into a unified framework.”

2. Introduction

• The introduction is detailed but too long and descriptive, reading more like a literature review than a problem statement.

• The research gap is stated in general terms (lack of integrated frameworks) but not theoretically or empirically substantiated. The authors should demonstrate how their model advances behavioral finance theory beyond existing TPB–UTAUT integrations already seen in recent work (e.g., Bouteraa, 2024; Hassan et al., 2024).

• The motivation for focusing on Vietnam is clear but underdeveloped; comparative insights or justification for cultural relevance (e.g., collectivist society, crypto regulation) would strengthen the contextual contribution.

Responses:

Thank you for your comments.

In the revised version, the Introduction section has been shortened and focused on the research problem, theoretical gaps and main contributions.

The gap has been clarified with specific evidence from previous studies, and at the same time, highlighting how this study extends behavioral finance theory through the integration of impulsivity and fintech self-efficacy.

The Vietnamese context section has also been added, emphasizing the characteristics of collective culture, the influence of social networks and the incomplete legal environment.

3. Theoretical Framework and Hypotheses

• The integration of four theories (UTAUT, TPB, Self-efficacy, Behavioral Finance) is ambitious but insufficiently justified. The paper lists theories rather than explaining how their constructs conceptually interlock.

• Some hypotheses are mechanical restatements of prior literature rather than theory-driven. For example:

o H5 (Impulsivity → Investment Decision) lacks novelty and empirical justification for cryptocurrency specifically.

o H8 (moderation by Fintech Self-Efficacy) should include theoretical rationale—why does self-efficacy strengthen, not weaken, the attitude–behavior link?

• Several hypotheses (H9–H11) use mediation/moderation terms interchangeably without explicit conceptual diagrams or theoretical logic for directionality.

Responses:

Thank you for your valuable comments.

In the revision, the author clarified the theoretical basis for integrating the four theories, which is also presented with the limitations of each theory in the context of cryptocurrencies.

The hypotheses have been adjusted in a theoretical direction.

For H5, the author has added an argument about the specificity of impulsive behavior in the context of cryptocurrencies.

Hypothesis H8 is extended with the theoretical basis of Bandura (1977) to explain the enhancing role of fintech self-efficacy on the attitude–behavior relationship.

In addition, the mediation and moderation hypotheses (H9–H11) have been more clearly distinguished, adding an explanation of the impact arrows in the research model.

4. Methodology

• Measurement scales are adopted from diverse sources, but no evidence of translation process, or pilot testing is provided. Reliability is reported (Cronbach’s alpha), yet discriminant validity is only briefly assessed (HTMT table) but the discussion seems as Pearson correlation discussion.

• The study uses PLS-SEM, which is suitable for exploratory models; however, justification for choosing PLS over covariance-based SEM is not provided. The authors should explain why PLS was used (e.g., non-normal data, model complexity).

• Common method bias (CMB) is likely, as all data are self-reported. The paper should apply Harman’s single-factor test or VIF statistics to mitigate this concern.

• Multivariate normality is not assessed.

Responses:

Thanks for your comment.

The author has revised and explained in detail the discriminant validity test, based on the HTMT index as recommended by Henseler et al. (2015).

The author also tested the common method variance (CMB) using Harman’s single-factor test and VIF, and the results showed that there were no serious problems with CMB.

Multivariate normality test was also added to ensure the reliability of the data before conducting the structural model analysis.

5. Results and Interpretation

• All eleven hypotheses are supported, which is statistically improbable and suggests potential model overfitting or confirmation bias.

• confidence intervals should be reported

• The f² and Q² table is mislabelled (“Bảng” indicates incomplete editing). Effect sizes are inconsistently interpreted—particularly for the mediated relationship, are described without backing.

• The mediation results (H9–H11) are not shown through indirect path coefficients or confidence intervals; these are essential for transparency.

• The interpretation remains descriptive, repeating numerical findings without deeper behavioral or cultural interpretation (e.g., why impulsivity still exerts a positive effect in Vietnam).

Responses:

Thank you for your comments.

Due to the need to modify some scales, the survey results with these new scales have a statistically insignificant hypothesis 7.

The author added confidence intervals for each relationship, and clarified the indirect path coefficients and the results of the mediation test (H9 - H11).

The f² and Q² tables have been labeled and interpreted consistently according to Cohen's threshold, and additional content to evaluate the results.

The results and interpretation section has been expanded to further explain the behavior and cultural context of Vietnamese investors, especially explaining why Impulsivity still has a positive impact in the high-risk crypto investment environment where psychological factors and social influences play a strong role.

6. Discussion and Contributions

• The discussion section repeats the results rather than linking them to broader behavioral finance or fintech literature.

• The claimed theoretical contributions (extending TPB, UTAUT, self-efficacy) are not demonstrated empirically; instead, they remain declarative.

• Practical implications are extensive but lack prioritization or feasibility analysis (e.g., “regulatory oversight” and “public education campaigns” are policy-level actions requiring evidence-based design).

Responses:

Thank you for your valuable comments.

The author has adjusted the discussion content to be more closely linked to the theoretical foundation of behavioral finance and technology acceptance, clarifying how psychological, social and technological factors interact in cryptocurrency investment behavior.

Theoretical contributions are concretized through the mediating role of investment attitudes and the moderating role of FinTech self-efficacy.

In terms of practical implications, the study has added the priority and feasibility section, focusing on specific measures such as financial education programs, FinTech training instead of just general policy recommendations.

7. Limitations and Future Research

• The limitations are appropriately listed (self-report, cross-sectional, young sample), but no mitigation strategies are suggested. Including remedies (e.g., longitudinal or multi-source data) would enhance credibility.

• Future research directions should identify specific constructs or moderating variables (e.g., financial regulation awareness, perceived risk, or crypto trust) rather than restating standard methodological fixes.

Responses:

Thank you for your comments.

The study added measures to overcome the limitations, such as proposing to incorporate real life behavioral data, expanding the research sample, and applying a longitudinal research design to increase reliability and generalizability. In addition, specific future research directions such as financial regulatory awareness, risk perception, and trust in cryptocurrency platforms are proposed to strengthen the theoretical and practical value.

8. Writing Quality and Formatting

• The manuscript is mostly well-written but shows signs of AI-assisted phrasing—overly polished transitions (“Taken together,” “This study enriches,” etc.) and uniform sentence rhythm.

• Some references lack full details or formatting consistency.

• Figures and tables are not numbered or cross-referenced properly (e.g., “Fig 1” lacks caption clarity)

Responses:

Thank you for your comments.

The entire article has been revised to ensure more natural and diverse sentence structure and transitions.

The list of references has been reviewed and revised.

In addition, tables and figures have been numbered, with additional titles and clear references in the article content.

9. Ethical and Data Transparency

• The data availability statement (“available on request”) conflicts with PLOS ONE’s open-data policy. Data should be deposited in a public repository with anonymized files unless privacy law prohibits it.

• The AI acknowledgment is transparent but should include the exact purpose (“used for grammar improvement only”).

Responses:

Thanks for your guidance.

The research data will be stored in accordance with PLOS ONE policy, with the files anonymized.

The AI acknowledgment has been edited to clarify that ChatGPT was used only to improve grammar during the manuscript editing process

---

## [Decision Letter · Decision Letter 1]

26 Nov 2025

We look forward to receiving your revised manuscript.

Kind regards,

Nishi Malhotra

Academic Editor

PLOS ONE

Journal Requirements:

Reviewers' comments:

Reviewer's Responses to Questions

**Comments to the Author**

Reviewer #3: All comments have been addressed

Reviewer #4: (No Response)

2. Is the manuscript technically sound, and do the data support the conclusions?

Reviewer #3: Yes

Reviewer #4: Yes

3. Has the statistical analysis been performed appropriately and rigorously?

Reviewer #3: (No Response)

Reviewer #4: Yes

4. Have the authors made all data underlying the findings in their manuscript fully available?

Reviewer #3: Yes

Reviewer #4: Yes

5. Is the manuscript presented in an intelligible fashion and written in standard English?

Reviewer #3: Yes

Reviewer #4: Yes

Reviewer #3: The current policy implications are too descriptive and broad. They primarily emphasize the significance of education, regulation, and protections based on platforms, but they lack the comprehensive cross-sector framework needed to manage a complicated digital-financial ecosystem. To enhance this section, the authors ought to integrate perspectives from three essential works—Steif’s Public Policy Analytics, Weimer & Vining’s Policy Analysis: Concepts and Practice, and Gohwong’s (2021) comparative analysis of crypto governance. These sources aid in defining a more transparent cross-sector rationale that links regulators, banks, fintech companies, investors, universities, and digital platforms, allowing the policy implications to progress from broad suggestions to a consistent, multi-actor governance framework.

The existing policy ramifications continue to be too broad and complicated. The proposed policies represent extensive technocratic overreach grounded in contemporary policy models. The need for digital-financial education using evidence-driven targeting and behavioral insights is patronizing, intending to manipulate investor behavior instead of fostering independent decision-making. This paternalism encompasses universities, which are anticipated to mandate simulation-based training, as well as banks and fintech companies, which are required to transform their apps into digital guardians with obligatory risk assessments and expense monitoring. Additionally, the drive for adaptive oversight in monitoring influencers is merely a surveillance directive, and the government's ultimate effort to reshape the national narrative is a cynical endeavor to substitute authentic market sentiment with government-sanctioned data-driven reasoning.

References

Gohwong, S. (2021). Comparative non-government crypto policy between Thailand and Argentina: A lesson from different public policy policy. Journal of Government and Politics, 12(3), 367–392.

Steif, K. (2022). Public policy analytics: Code and context for data science in government.

CRC Press.

Weimer, D. L., & Vining, A. R. (2017). Policy analysis: Concepts and practice (6th ed.).

Routledge.

Note: Suggested improvements for completeness (optional if overly burdensome)

Reviewer #4: Several minor issues remain that should be addressed to further enhance clarity and publication-worthiness:

Digital Financial Literacy (DFL) Measurement

The revision of the Digital Financial Literacy scale is appreciated. However, item FL4 (“I feel that my knowledge and expertise will enable me to outperform the market”) does not measure digital literacy. Instead, it reflects confidence or overconfidence, which is conceptually distinct from literacy. Replacing or rewording this item to focus on knowledge or ability. For example, the ability to evaluate platform legitimacy or digital security would improve construct validity.

Behavioral Finance Framework

The discussion of Behavioral Finance is still somewhat narrow and remains framed primarily around impulsivity. Behavioral Finance encompasses a broader set of well-established cognitive and emotional biases including overconfidence, herding, anchoring, and loss aversion which should be briefly acknowledged to provide a more complete theoretical grounding.

FinTech-Related Risk Mechanisms

While the manuscript refers to FinTech’s role in facilitating cryptocurrency investment and increasing risk exposure, the explanation of the underlying mechanism requires greater detail. Expanding this section to describe how digital platforms reduce friction, enable rapid decision-making, and amplify risk-taking would strengthen the argument.

Impulsivity Justification

Although the definitions have been clarified, the rationale for including impulsivity as a predictor remains somewhat limited. Consider strengthening this section with references to behavioral mechanisms (e.g., sensation seeking, delay discounting) or contextual factors that make impulsivity particularly relevant to crypto investment behavior.

Item Wording and Clarity

A few measurement items still include metaphorical or culturally specific phrasing that may introduce interpretive ambiguity. Ensuring that all items are phrased in clear, direct, and culturally neutral language will improve response reliability. For example, bungee jumping and thrilling ride.

Writing Style and Flow

Certain sections exhibit uniform sentence rhythm and transition patterns that give the appearance of AI-assisted phrasing. A light manual edit to vary sentence structure and transitions would enhance readability.

**Do you want your identity to be public for this peer review?** For information about this choice, including consent withdrawal, please see our For information about this choice, including consent withdrawal, please see our Privacy Policy .

Reviewer #3: **Yes:** Srirath GohwongSrirath Gohwong

Reviewer #4: No

---

## [Author Response · Author response to Decision Letter 2]

3 Dec 2025

I would like to express my sincere gratitude to the reviewers for their insightful and constructive comments. In the revised manuscript, all modifications made in response to the reviewers’ suggestions.

Review #3

Review Comments to the Author

The current policy implications are too descriptive and broad. They primarily emphasize the significance of education, regulation, and protections based on platforms, but they lack the comprehensive cross-sector framework needed to manage a complicated digital-financial ecosystem. To enhance this section, the authors ought to integrate perspectives from three essential works—Steif’s Public Policy Analytics, Weimer & Vining’s Policy Analysis: Concepts and Practice, and Gohwong’s (2021) comparative analysis of crypto governance. These sources aid in defining a more transparent cross-sector rationale that links regulators, banks, fintech companies, investors, universities, and digital platforms, allowing the policy implications to progress from broad suggestions to a consistent, multi-actor governance framework.

The existing policy ramifications continue to be too broad and complicated. The proposed policies represent extensive technocratic overreach grounded in contemporary policy models. The need for digital-financial education using evidence-driven targeting and behavioral insights is patronizing, intending to manipulate investor behavior instead of fostering independent decision-making. This paternalism encompasses universities, which are anticipated to mandate simulation-based training, as well as banks and fintech companies, which are required to transform their apps into digital guardians with obligatory risk assessments and expense monitoring. Additionally, the drive for adaptive oversight in monitoring influencers is merely a surveillance directive, and the government's ultimate effort to reshape the national narrative is a cynical endeavor to substitute authentic market sentiment with government-sanctioned data-driven reasoning.

References

Gohwong, S. (2021). Comparative non-government crypto policy between Thailand and Argentina: A lesson from different public policy policy. Journal of Government and Politics, 12(3), 367–392.

Steif, K. (2022). Public policy analytics: Code and context for data science in government. CRC Press.

Weimer, D. L., & Vining, A. R. (2017). Policy analysis: Concepts and practice (6th ed.). Routledge.

Note: Suggested improvements for completeness (optional if overly burdensome)

Responses

Based on the suggestions, the authors revised the policy implications, specifically as follows (from line 595 to line 616):

- The policy implications section has been adjusted to directly respond to criticisms of technocratic intervention and paternalism in previous proposals.

- The revision removes mandatory measures and avoids behavioral guidance, instead emphasizing voluntariness, transparency, and respect for investor autonomy.

- Digital financial education is presented as a support tool, not a mechanism for behavioral manipulation.

The role of universities, banks, and fintech companies is shifted from “guardians” to optional providers, leaving users to decide their own level of participation.

- Influencer supervision is reinterpreted as limited and proportionate, avoiding the perception of excessive supervision.

Review #4

Review Comments to the Author

Digital Financial Literacy (DFL) Measurement

The revision of the Digital Financial Literacy scale is appreciated. However, item FL4 (“I feel that my knowledge and expertise will enable me to outperform the market”) does not measure digital literacy. Instead, it reflects confidence or overconfidence, which is conceptually distinct from literacy. Replacing or rewording this item to focus on knowledge or ability. For example, the ability to evaluate platform legitimacy or digital security would improve construct validity.

Responses

Item FL4 has been revised to: “I can use my knowledge to evaluate the legitimacy, risks, and security features of cryptocurrency platforms before making investment decisions” (line 433).

Review Comments to the Author

Behavioral Finance Framework

The discussion of Behavioral Finance is still somewhat narrow and remains framed primarily around impulsivity. Behavioral Finance encompasses a broader set of well-established cognitive and emotional biases including overconfidence, herding, anchoring, and loss aversion which should be briefly acknowledged to provide a more complete theoretical grounding.

Responses

In the revised version, the Behavioral Finance section has been expanded to more fully reflect cognitive and emotional biases. The author has added and explained important biases such as overconfidence, herding, anchoring, and loss aversion, instead of focusing only on impulsivity. Impulsivity is retained as a specific bias in the research context (from line 120 to line 129).

Review Comments to the Author

FinTech-Related Risk Mechanisms

While the manuscript refers to FinTech’s role in facilitating cryptocurrency investment and increasing risk exposure, the explanation of the underlying mechanism requires greater detail. Expanding this section to describe how digital platforms reduce friction, enable rapid decision-making, and amplify risk-taking would strengthen the argument.

Responses

The author has revised and expanded this section according to reviewer comments, explaining more clearly the mechanism of FinTech digital platforms that reduce transaction friction, support quick decision-making, and increase investors' risk tolerance (from line 301 to line 308).

Review Comments to the Author

Impulsivity Justification

Although the definitions have been clarified, the rationale for including impulsivity as a predictor remains somewhat limited. Consider strengthening this section with references to behavioral mechanisms (e.g., sensation seeking, delay discounting) or contextual factors that make impulsivity particularly relevant to crypto investment behavior.

Responses

The author revised the theory of impulsivity according to the reviewer’s comments. Behavioral mechanisms such as sensation seeking and delay discounting were added. The revision also emphasizes specific contextual factors of the cryptocurrency market. These include the FOMO effect and the desire for quick profits (from line 145 to line 154).

Review Comments to the Author

Item Wording and Clarity

A few measurement items still include metaphorical or culturally specific phrasing that may introduce interpretive ambiguity. Ensuring that all items are phrased in clear, direct, and culturally neutral language will improve response reliability. For example, bungee jumping and thrilling ride.

Responses

The author has reviewed the scales and adjusted them according to comments (line 433).

Review Comments to the Author

Writing Style and Flow

Certain sections exhibit uniform sentence rhythm and transition patterns that give the appearance of AI-assisted phrasing. A light manual edit to vary sentence structure and transitions would enhance readability.

Responses

The author has reviewed and edited the entire article.

---

## [Decision Letter · Decision Letter 2]

10 Jan 2026

Dear Dr.  Tang,

Thank you for submitting your manuscript to PLOS ONE. After careful consideration, we feel that it has merit but does not fully meet PLOS ONE’s publication criteria as it currently stands. Therefore, we invite you to submit a revised version of the manuscript that addresses the points raised during the review process.

We look forward to receiving your revised manuscript.

Kind regards,

Nishi Malhotra

Academic Editor

PLOS One

Journal Requirements:

Reviewer's Responses to Questions

**Comments to the Author**

Reviewer #5: (No Response)

Reviewer #6: All comments have been addressed

2. Is the manuscript technically sound, and do the data support the conclusions?

Reviewer #5: Yes

Reviewer #6: Yes

3. Has the statistical analysis been performed appropriately and rigorously?

Reviewer #5: No

Reviewer #6: Yes

4. Have the authors made all data underlying the findings in their manuscript fully available?

Reviewer #5: Yes

Reviewer #6: Yes

5. Is the manuscript presented in an intelligible fashion and written in standard English?

Reviewer #5: Yes

Reviewer #6: Yes

Reviewer #5: The manuscript offers a potentially valuable integration of UTAUT, TPB, and behavioral finance to study cryptocurrency investment decisions in Vietnam. However, several material inconsistencies in statistical reporting and interpretation currently limit the evidentiary strength of the claims. The issues below should be addressed：

1. Table 5 contains internal inconsistencies. For H2 and H8, statistically significant p-values (p < 0.05) are reported alongside 95% confidence intervals that include zero (e.g., H8: [−0.062, 0.143]). Under standard two-sided inference, these statements are not jointly compatible and suggest either a transcription/labeling problem (e.g., the interval is not the stated CI) or an error in reporting.

2. In addition, H4 and H5 display identical point estimates and t-statistics (β = 0.476, t = 11.031), which raises the possibility of a mistake. The authors should conduct a rigorous audit to ensure the accuracy, validityof the reported statistical results.

3. the Abstract is inconsistent with H1. It states that digital financial literacy reduces impulsivity, whereas Table 5 reports a strong positive effect (β = 0.729, p < 0.001). The current explanation is difficult to reconcile with items such as acting “on the spur of the moment” or “without deliberation,” which capture rashness rather than informed decisiveness. The authors should provide a theory-consistent mechanism (e.g., overconfidence, illusion of control, or social-media-driven perceived literacy) and distinguish rapid informed decisions from affect-driven impulsivity.

4. the policy implications should reflect H1. If higher digital financial literacy predicts higher impulsivity, recommendations focused on expanding technical literacy may be incomplete and could exacerbate impulsive trading without complementary behavioral training. The implications should explicitly emphasize behavioral discipline and emotional regulation alongside technical skills.

5. H8 is interpreted inconsistently with its sign. The interaction coefficient is negative (β = −0.079), yet the text claims a strengthening effect. A negative interaction implies attenuation. Subject to corrected reporting, the authors should revise the interpretation and state the practical meaning of the negative moderation.

Reviewer #6: The authors have adequately addressed all comments raised in the previous round of review. The revised manuscript demonstrates improved conceptual clarity, stronger theoretical grounding, and clearer justification of key constructs, particularly regarding digital financial literacy, impulsivity, and fintech self-efficacy.

The methodology is technically sound, the sample size is appropriate, and the statistical analysis using PLS-SEM is rigorous and well reported. The results support the conclusions drawn, and the discussion appropriately interprets both direct and indirect effects without overstatement.

Minor language and formatting issues are minimal and do not affect the scientific quality of the work. Overall, the manuscript is clearly written, coherent, and suitable for publication in its current form.

**Do you want your identity to be public for this peer review?** For information about this choice, including consent withdrawal, please see our For information about this choice, including consent withdrawal, please see our Privacy Policy .

Reviewer #5: No

Reviewer #6: **Yes:** Prof.Amjed Abbas AhmedProf.Amjed Abbas Ahmed

Regards

Dr Nishi Malhotra

NAAS will assess whether your figures meet our technical requirements by comparing each figure against our figure specifications

---

## [Author Response · Author response to Decision Letter 3]

15 Jan 2026

I would like to express my sincere gratitude to the reviewers for their insightful and constructive comments. In the revised manuscript, all modifications made in response to the reviewers’ suggestions.

Review #5

Review Comments to the Author

1. Table 5 contains internal inconsistencies. For H2 and H8, statistically significant p-values (p < 0.05) are reported alongside 95% confidence intervals that include zero (e.g., H8: [−0.062, 0.143]). Under standard two-sided inference, these statements are not jointly compatible and suggest either a transcription/labeling problem (e.g., the interval is not the stated CI) or an error in reporting.

Responses

Thank you for highlighting this issue. The author rechecked the bootstrapping output and found that the reported 95% confidence intervals for H2 and H8 in Table 5 contained transcription errors. The confidence interval values have now been corrected (Table 5).

Review #4

Review Comments to the Author

In addition, H4 and H5 display identical point estimates and t-statistics (β = 0.476, t = 11.031), which raises the possibility of a mistake. The authors should conduct a rigorous audit to ensure the accuracy, validity of the reported statistical results.

Responses

Thank you for pointing this out. The author audited the structural model results and identified a transcription error in Table 5, where the coefficients for H4 and H5 were inadvertently duplicated. The table has now been corrected (Table 5).

Review Comments to the Author

The Abstract is inconsistent with H1. It states that digital financial literacy reduces impulsivity, whereas Table 5 reports a strong positive effect (β = 0.729, p < 0.001). The current explanation is difficult to reconcile with items such as acting “on the spur of the moment” or “without deliberation,” which capture rashness rather than informed decisiveness. The authors should provide a theory-consistent mechanism (e.g., overconfidence, illusion of control, or social-media-driven perceived literacy) and distinguish rapid informed decisions from affect-driven impulsivity.

Responses

Thank you for this insightful comment. The author revised the Abstract (Line 14-25) and Discussion (line 425-435) to ensure consistency with H1 and the empirical results reported in Table 5. The revised manuscript no longer states that digital financial literacy reduces impulsivity. Instead, it clarifies that higher digital financial literacy is positively associated with impulsivity related investment behavior in the cryptocurrency context

The author also explicitly distinguish rapid informed decision making from affect driven impulsivity by emphasizing that impulsivity is measured through spontaneous and minimally deliberated actions, which capture emotional rashness rather than informed decisiveness. Drawing on behavioral finance theory, this positive relationship is explained through overconfidence and illusion of control, reinforced by perceived digital competence and social media exposure in highly volatile markets.

Review Comments to the Author

The policy implications should reflect H1. If higher digital financial literacy predicts higher impulsivity, recommendations focused on expanding technical literacy may be incomplete and could exacerbate impulsive trading without complementary behavioral training. The implications should explicitly emphasize behavioral discipline and emotional regulation alongside technical skills.

Responses

Thank you for instruction. The author has revised the Policy Implications, emphasizing the integration of behavioral discipline and emotional regulation alongside digital financial skills to mitigate impulsive trading and promote sustainable investment behavior (line 527-537; line 553-560).

Review Comments to the Author

H8 is interpreted inconsistently with its sign. The interaction coefficient is negative (β = −0.079), yet the text claims a strengthening effect. A negative interaction implies attenuation. Subject to corrected reporting, the authors should revise the interpretation and state the practical meaning of the negative moderation.

Responses

Thank you for noting this issue. The interaction coefficient for H8 was incorrectly reported during result rewriting and has been corrected to a positive value (β = 0.079) in the revised manuscript. The interpretation has been revised accordingly to reflect a small positive moderating effect and its practical meaning (line 404 – 410).

Review #6

Minor language and formatting issues are minimal and do not affect the scientific quality of the work. Overall, the manuscript is clearly written, coherent, and suitable for publication in its current form.

Responses

Thank you very much for this encouraging comment. The author has continued to review the manuscript and made minor language and formatting adjustments (all lines)

---

## [Decision Letter · Decision Letter 3]

5 Feb 2026

Dear Dr. Tang,

Thank you for submitting your manuscript to PLOS ONE. After careful consideration, we feel that it has merit but does not fully meet PLOS ONE’s publication criteria as it currently stands. Therefore, we invite you to submit a revised version of the manuscript that addresses the points raised during the review process.

We look forward to receiving your revised manuscript.

Kind regards,

Nishi Malhotra

Academic Editor

PLOS One

Journal Requirements:

Reviewers' comments:

Reviewer's Responses to Questions

**Comments to the Author**

Reviewer #5: All comments have been addressed

Reviewer #7: (No Response)

2. Is the manuscript technically sound, and do the data support the conclusions?

Reviewer #5: Partly

Reviewer #7: Yes

3. Has the statistical analysis been performed appropriately and rigorously?

Reviewer #5: I Don't Know

Reviewer #7: Yes

4. Have the authors made all data underlying the findings in their manuscript fully available?

Reviewer #5: No

Reviewer #7: (No Response)

5. Is the manuscript presented in an intelligible fashion and written in standard English?

Reviewer #5: Yes

Reviewer #7: Yes

Reviewer #5: The author has addressed all the comments, but the authenticity of the empirical results remains a cause for concern. Specifically, the coefficient for H8 was inverted from negative to positive without any corresponding change in the t-value or p-value, which is statistically implausible. Furthermore, regarding H5, while the path coefficient and t-value were updated, the confidence interval remains unchanged, suggesting a lack of rigorous re-analysis. These discrepancies cast significant doubt on the authenticity and reliability of the experimental results. Therefore, I recommend that the manuscript not be accepted until the authors provide the original software output logs (e.g., SmartPLS report) to substantiate the reported findings.

Reviewer #7: PONE-D-25-46593R3

I have carefully reviewed the revised version of the manuscript, and I have three main comments that must be discussed before its publication: While they are minor, they are essential.

1. First, financial literacy teaches us rational investing, motivated by at least three main analyses: fundamental analysis, cyclical effects, and technical analysis. However, the crypto market is dominated by retail investors and weak fundamentals. Even if they do not exhibit specific cyclical effects that can be exploited using technical or other analysis. Thus, please explain this mechanism in a clearer fashion: How financial literacy motivates investing in cryptos. Perhaps some literature support comes from Ali et al. (2025). https://doi.org/10.1186/s40854-024-00686-4.

2. Second, beyond crypto investing, the authors should discuss investing preferences in the Asian markets, as they have studied the Vietnamese market. In this respect, Ülkü et al. (2023) offer a comprehensive understanding of investing habitat I the Asian markets. It will help us to understand the financial decisions made by the investors, specifically retail and foreign investors, which may help the authors to enhance the clarity and generalizability of this study. https://doi.org/10.1016/j.pacfin.2023.102044.

Once these two aspects are discussed, the manuscript can be accepted as it is in good shape after three rounds of revision. Overall, it's an interesting paper and I recommend its publication.

Good luck!

**Do you want your identity to be public for this peer review?** For information about this choice, including consent withdrawal, please see our For information about this choice, including consent withdrawal, please see our Privacy Policy .

Reviewer #5: No

Reviewer #7:**Yes:** Fahad AliFahad Ali

---

## [Author Response · Author response to Decision Letter 4]

11 Feb 2026

I would like to express my sincere gratitude to the reviewers for their insightful and constructive comments. In the revised manuscript, all modifications made in response to the reviewers’ suggestions.

Review #5

Review Comments to the Author

1. The author has addressed all the comments, but the authenticity of the empirical results remains a cause for concern. Specifically, the coefficient for H8 was inverted from negative to positive without any corresponding change in the t-value or p-value, which is statistically implausible. Furthermore, regarding H5, while the path coefficient and t-value were updated, the confidence interval remains unchanged, suggesting a lack of rigorous re-analysis. These discrepancies cast significant doubt on the authenticity and reliability of the experimental results. Therefore, I recommend that the manuscript not be accepted until the authors provide the original software output logs (e.g., SmartPLS report) to substantiate the reported findings.

Responses

Thank you for your guidance. In the revised version, the author re-estimated the full PLS-SEM model, using one bootstrapping procedure with 5,000 resamples, and all results reported in Table 5. The moderating effect in H8 is now negative and statistically significant (β = –0.082, t = 2.140, p = 0.032; CI = [–0.154; –0.001]), and the sign and test statistics are fully consistent. For H5, the confidence interval was recomputed after re-running the model and now matches the updated coefficient and t-value (β = 0.163, t = 2.212, p = 0.027; CI = [0.017; 0.306]). All reported values are taken directly from the SmartPLS output.

Review #7

Review Comments to the Author

I have carefully reviewed the revised version of the manuscript, and I have three main comments that must be discussed before its publication: While they are minor, they are essential.

1. First, financial literacy teaches us rational investing, motivated by at least three main analyses: fundamental analysis, cyclical effects, and technical analysis. However, the crypto market is dominated by retail investors and weak fundamentals. Even if they do not exhibit specific cyclical effects that can be exploited using technical or other analysis. Thus, please explain this mechanism in a clearer fashion: How financial literacy motivates investing in cryptos. Perhaps some literature support comes from Ali et al. (2025). https://doi.org/10.1186/s40854-024-00686-4.

Responses

Thank you for this comment. The author revised the manuscript to clarify that, in cryptocurrency markets, financial literacy does not motivate investment through traditional fundamental, cyclical, or technical analysis. Instead, it supports informed participation by improving investors’ understanding of market volatility, risk exposure, and the practical use of trading platforms, as well as their ability to manage risk through diversification and hedging. The theoretical explanation has been adjusted accordingly, and recent literature has been used to support this revised mechanism.

Review Comments to the Author

Second, beyond crypto investing, the authors should discuss investing preferences in the Asian markets, as they have studied the Vietnamese market. In this respect, Ülkü et al. (2023) offer a comprehensive understanding of investing habitat I the Asian markets. It will help us to understand the financial decisions made by the investors, specifically retail and foreign investors, which may help the authors to enhance the clarity and generalizability of this study. https://doi.org/10.1016/j.pacfin.2023.102044.

Once these two aspects are discussed, the manuscript can be accepted as it is in good shape after three rounds of revision. Overall, it's an interesting paper and I recommend its publication.

Responses

Thank you for this suggestion. The author added a brief discussion in the Discussion section to situate the Vietnamese cryptocurrency market within the broader context of Asian financial markets. Specifically, the discussion highlights that retail investors dominate trading activity, tend to trade more actively during periods of uncertainty, and behave differently from foreign institutional investors, based on evidence from Asian markets.

---

## [Decision Letter · Decision Letter 4]

26 Feb 2026

We look forward to receiving your revised manuscript.

Kind regards,

Nishi Malhotra

Academic Editor

PLOS One

Journal Requirements:

Reviewers' comments:

Reviewer's Responses to Questions

**Comments to the Author**

Reviewer #1: (No Response)

Reviewer #7: All comments have been addressed

2. Is the manuscript technically sound, and do the data support the conclusions?

Reviewer #1: Yes

Reviewer #7: Yes

3. Has the statistical analysis been performed appropriately and rigorously?

Reviewer #1: N/A

Reviewer #7: Yes

4. Have the authors made all data underlying the findings in their manuscript fully available?

Reviewer #1: No

Reviewer #7: Yes

5. Is the manuscript presented in an intelligible fashion and written in standard English?

Reviewer #1: Yes

Reviewer #7: Yes

Reviewer #1: I share a previous referee concerns about the authenticity of the empirical results. The discrepancies previously found cast significant doubt on the authenticity and reliability of the experimental results. The authors have fixed them but not provided output logs as requested by a previous referee. Therefore, I also recommend that the manuscript not be accepted until the authors provide the original software output logs (e.g., SmartPLS report) to substantiate the reported findings. If they do, my recommendation would be to accept, as the paper is now otherwise ready.

Reviewer #7: The manuscript is substantially improved and can be considered for publication in PLOS One. I have no more comments. Good luck!

**Do you want your identity to be public for this peer review?** For information about this choice, including consent withdrawal, please see our For information about this choice, including consent withdrawal, please see our Privacy Policy .

Reviewer #1: No

Reviewer #7: No

---

## [Author Response · Author response to Decision Letter 5]

1 Mar 2026

Thank you for raising this concern. I have provided three original SmartPLS output files as supplementary materials. These files include the full measurement model results, structural model results, and bootstrapping outputs.

In the final revision, I rechecked all coefficients, p-values, confidence intervals, and model indicators (R², f², Q²) against the original software reports. A few minor corrections were made to ensure that the numbers and interpretations in the manuscript match the SmartPLS outputs exactly. These revisions relate only to presentation and wording. There were no changes to the dataset, the model specification, or the estimation procedure. I sincerely apologize for any confusion these inconsistencies may have caused and respectfully ask for your understanding regarding these final adjustments. The revisions were made solely to improve reporting accuracy and ensure full consistency between the manuscript and the original analytical results.

---

## [Decision Letter · Decision Letter 5]

4 Mar 2026

Integrating Behavioral, Social, and Technological Factors in Cryptocurrency Investment Decisions

PONE-D-25-46593R5

Dear Dr. Sang Tang

We’re pleased to inform you that your manuscript has been judged scientifically suitable for publication and will be formally accepted for publication once it meets all outstanding technical requirements.

Kind regards,

Nishi Malhotra

Academic Editor

PLOS One

Additional Editor Comments (optional):

Reviewers' comments:

Reviewer's Responses to Questions

**Comments to the Author**

Reviewer #1: All comments have been addressed

Reviewer #7: All comments have been addressed

2. Is the manuscript technically sound, and do the data support the conclusions?

Reviewer #1: Yes

Reviewer #7: Yes

3. Has the statistical analysis been performed appropriately and rigorously?

Reviewer #1: Yes

Reviewer #7: Yes

4. Have the authors made all data underlying the findings in their manuscript fully available?

Reviewer #1: No

Reviewer #7: Yes

5. Is the manuscript presented in an intelligible fashion and written in standard English?

Reviewer #1: Yes

Reviewer #7: Yes

Reviewer #1: After viewing the logs, I am satisfied and have no further comments. The manuscript is ready for publication.

Reviewer #7: My comments have been addressed. The manuscript is substantially improved and can be accepted for publication.

**Do you want your identity to be public for this peer review?** For information about this choice, including consent withdrawal, please see our For information about this choice, including consent withdrawal, please see our Privacy Policy .

Reviewer #1: No

Reviewer #7: No

---

## [Editor Report · Acceptance letter]

PONE-D-25-46593R5

PLOS One

Dear Dr. Tang,

I'm pleased to inform you that your manuscript has been deemed suitable for publication in PLOS One. Congratulations! Your manuscript is now being handed over to our production team.

Kind regards,

on behalf of

Dr. Nishi Malhotra

Academic Editor

PLOS One